# Onion-like multicolor thermally activated delayed fluorescent carbon quantum dots for efficient electroluminescent light-emitting diodes

Yuxin Shi[1], Yang Zhang [ORCID][1] ✉, Zhibin Wang[2], Ting Yuan[1], Ting Meng[1], Yunchao Li [ORCID][1], Xiaohong Li[1], Fanglong Yuan [ORCID][1] ✉, Zhan'ao Tan [ORCID][3] ✉ & Louzhen Fan [ORCID][1] ✉

Carbon quantum dots are emerging as promising nanomaterials for next-generation displays. The elaborate structural design is crucial for achieving thermally activated delayed fluorescence, particularly for improving external quantum efficiency of electroluminescent light-emitting diodes. Here, we report the synthesis of onion-like multicolor thermally activated delayed fluorescence carbon quantum dots with quantum yields of 42.3–61.0%. Structural, spectroscopic characterization and computational studies reveal that onion-like structures assembled from monomer carbon quantum dots of different sizes account for the decreased singlet-triplet energy gap, thereby achieving efficient multicolor thermally activated delayed fluorescence. The devices exhibit maximum luminances of 3785–7550 cd m$^{-2}$ and maximum external quantum efficiency of 6.0–9.9%. Importantly, owing to the weak van der Waals interactions and adequate solution processability, flexible devices with a maximum luminance of 2554 cd m$^{-2}$ are realized. These findings facilitate the development of high-performance carbon quantum dots-based electroluminescent light-emitting diodes that are promising for practical applications.

Carbon quantum dots (CQDs) are considered promising nanomaterial candidates for optoelectronics[1–8] owing to their adjustable bandgap emission, high photo- and thermal stability, good solution processability, and low toxicity[9–12]. CQD-based light-emitting diodes (LEDs), in particular, hold considerable potential as a type of quantum-dot LEDs for next-generation, high-performance displays, comparable to that of organic LEDs (OLEDs)[13–18]. However, the external quantum efficiency (EQE) of electroluminescent LEDs based on fluorescence emission are limited by the electron charge spin (*S*) and merely harvests singlet

excitons (1/4 ratio, $S = 0$), which have a theoretical EQE limit of 5%[19–21]. Efficient utilization of triplet excitons (3/4 ratio, $S = 1$), either in the form of phosphorescence or delayed fluorescence, is expected to significantly improve device performance, including approaching an EQE$_{max}$ of nearly 25% and achieving high luminance[22–24].

More than 5% EQE for phosphorescent CQD-based electroluminescent LEDs was achieved in our lab by connecting pure CQDs with π-channel chains to obtain a two-dimensional (2D) framework. Reduced triplet energy levels, enhanced spin–orbit coupling (SOC),

---

[1]Key Laboratory of Theoretical & Computational Photochemistry of Ministry of Education, College of Chemistry, Beijing Normal University, Beijing 100875, China. [2]College of Physics and Energy, Fujian Normal University, Fuzhou 350117, China. [3]Beijing Advanced Innovation Centre for Soft Matter Science and Engineering, Beijing University of Chemical Technology, Beijing 100029, China. ✉e-mail: y.zhang@bnu.edu.cn; flyuan@bnu.edu.cn; tanzhanao@buct.edu.cn; lzfan@bnu.edu.cn

and balanced electron–hole pairs enabled the use of triplet excitons in CQDs for achieving high device efficiency[22]. However, phosphorescence emission originating from the large energy gap between singlet and triplet states ($\Delta E_{ST}$) commonly has one broad or two separate emission peaks[25,26]. Typically, these peaks exhibit a long lifetime ($\tau$), on the order of milliseconds and seconds, which are affected by spin-forbidden transitions[27,28].

Thermally activated delayed fluorescence (TADF), in which excitons experience a transition process from the triplet excited state ($T_1$) back to singlet excited state ($S_1$), and then to the ground state ($S_0$), has been a promising approach to overcome the aforementioned situation[29,30]. A value of $\Delta E_{ST}$ (<0.3 eV) has been widely reported as the key prerequisite for strong reverse intersystem crossing (RISC) through various strategies[31,32]. This can improve the charge transfer efficiency between energy levels, which will reduce the lifetime to the nanosecond and microsecond range, achieving full-color emission[33,34]. To date, carbon dots confined in zeolite have exhibited blue TADF with a quantum yield (QY) of 52.14% and ultralong lifetimes up to 350 ms[35]. However, CQDs with simple structural modifications still suffer from serious nonradiative recombination losses in the triplet state and have poor solution processability, hindering the practical application of TADF in electroluminescent LEDs.

In the case of OLEDs with rational molecular design, such as metal-coordination and donor–acceptor structures[36–39], $\Delta E_{ST}$ is proportional to the overlap between the highest occupied molecular orbital (HOMO) and the lowest unoccupied molecular orbital (LUMO)[40–42]. Therefore, structural assembly with specific molecular packing is an effective approach for regulating the energy levels of $S_1$ and $T_1$ states. Based on the synergetic interactions between the π–π conjugation and steric hindrance of the carbon core and surrounding organic molecules[14,43,44], each CQD exhibits a molecular geometry with steric distortion as a monomer. Furthermore, the structural assembly of monomer CQDs can form a specific three-dimensional (3D) spatial configuration, which can regulate the orbital overlap for narrowing $\Delta E_{ST}$ and realize TADF emission[45–47].

In this work, we synthesize onion-like multicolor TADF CQDs (OLM-TADF-CQDs). The monomer CQDs serve as building blocks composed of a carbon core ($\pi_{CQDs}$) and perylene-like molecules ($\pi_B$) with a 3D symmetrical BTC molecule as a rotatable phenyl bridge. Monomer CQDs are stacked by conjugating $\pi_{CQDs}$ and $\pi_B$ from adjacent CQDs, twisting them to generate a hexagon ring with a spatial corner angle ($\theta$) of 120°. With the continuous construction of monomer CQDs and hexagon rings in 3D stereo, the mono-layered structure is assembled, and a 3D onion-like structure is finally formed in a closed and nested manner. The OLM-TADF-CQDs with 3D self-assembled structures is expected to stabilize the triplet state and reduce the value of $\Delta E_{ST}$ owing to the small orbital overlap, resulting in blue (440 nm), green (490 nm), yellow (520 nm), orange (570 nm), and red (616 nm) TADF emissions with QYs up to 42.3–61.0%. Additionally, electroluminescent LEDs based on OLM-TADF-CQDs exhibit high stability and high performance with maximum luminances ($L_{max}$) of 3785–7550 cd m$^{-2}$ and maximum EQEs of 6.0–9.9%. Finally, the weak van der Waals interactions among the OLM-TADF-CQDs and adequate solution processability facilitate the fabrication of flexible LEDs based on red OLM-TADF-CQDs with a $L_{max}$ of 2554 cd m$^{-2}$, demonstrating high stability toward practical applications.

## Results

### Design and synthesis

The OLM-TADF-CQDs are synthesized by subjecting 1,3,5-benzene-tricarboxylic (BTC) acid, 3,4,9,10-perylenetetracarboxylic dianhydride, formamide, and concentrated sulfuric acid to solvothermal treatment. Through a series of purification processes, including water washing, dialysis, and silica gel column chromatography, pure samples are obtained. The reaction temperature is increased from 100 °C to 200 °C to tune their emission color, and the ratio of precursors is modulated to prepare monomer CQDs of different sizes. After cooling naturally to 300 K, OLM-TADF-CQDs with blue, green, yellow, orange, and red TADF emissions are synthesized; more details are presented in the "Method" section. Figure 1a represents the schematic of the synthesis

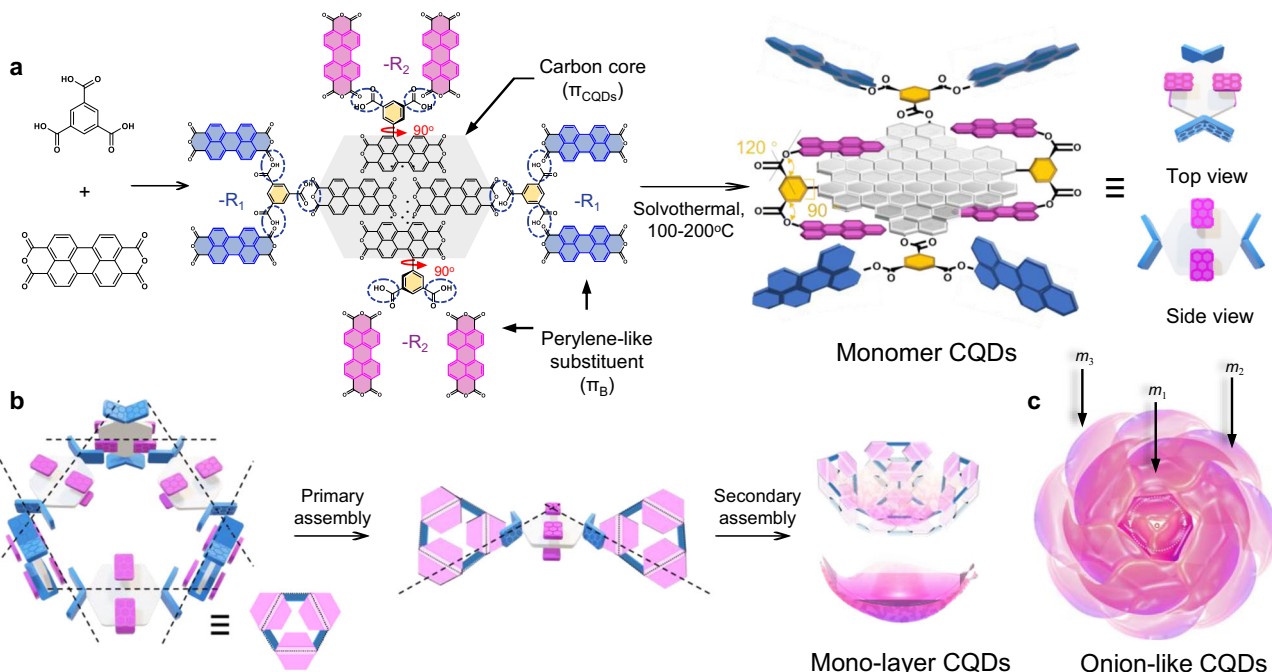

**Fig. 1 | Structure of OLM-TADF-CQDs. a** Synthesis route of monomer carbon quantum dots (CQDs) ($\pi_{CQDs}$: the carbon core colored as gray block, $\pi_B$: the perylene-like molecules colored as purple block (-R$_1$ with a $\theta$ of 120°) and blue block (-R$_2$ with a $\theta$ of 90°), a BTC molecule: the yellow block as a rotatable phenyl bridge (-90°)). **b** Assembly process and **c** the onion-like structure of onion-like multicolor thermally activated delayed fluorescence carbon quantum dots (OLM-TADF-CQDs) ($m_1$: the first one, $m_2$: the second one, and $m_3$: the third one).

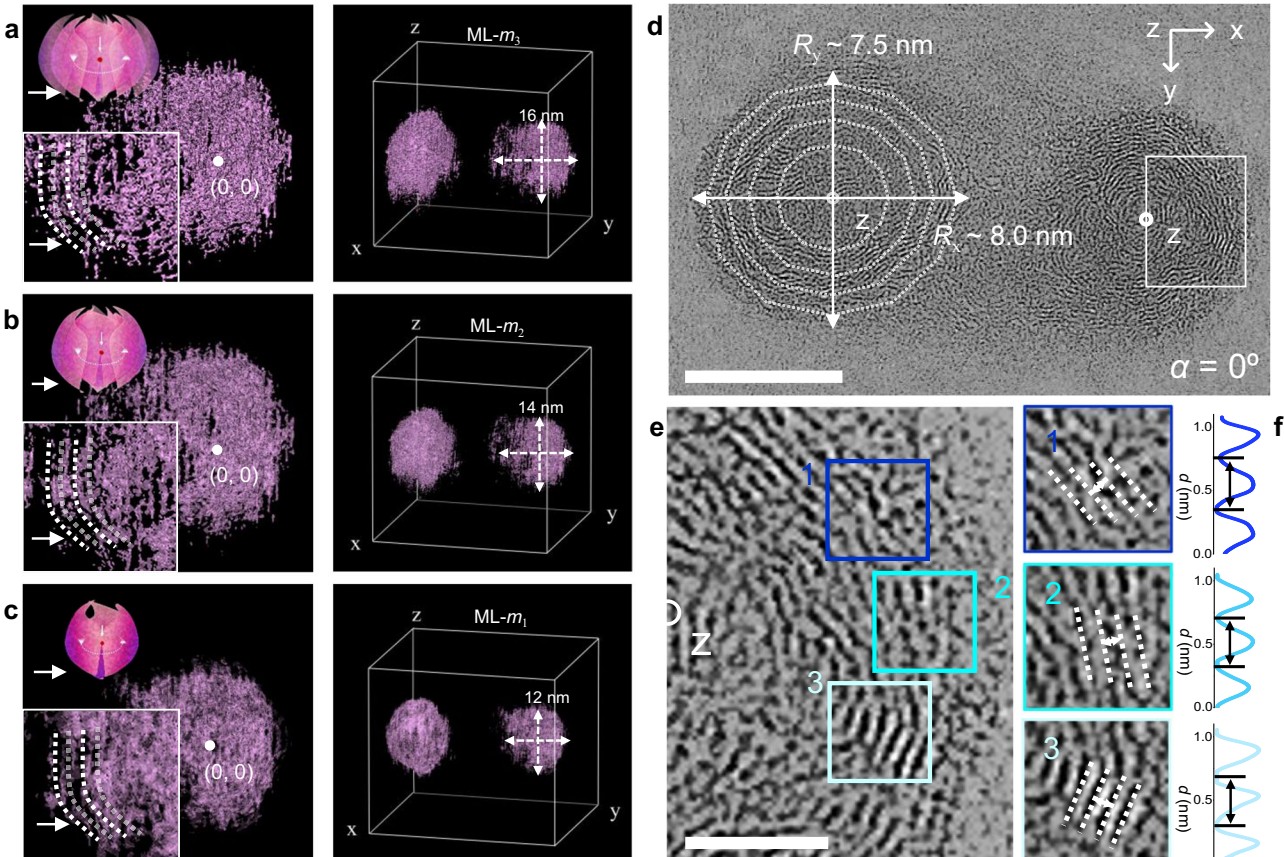

**Fig. 2 | Methodological approach for the 3D characterization.** 3D model and reconstruction images of the onion-like multi-layered structure by recording STEM tilt series (tilt angle: $\alpha$ from − 70° to + 70°) of multi-layered derivation process, starting from **a** ML $m_1$ (the first one of multi-layer), **b** ML $m_2$ (the second one of multi-layer), and **c** ML $m_3$ (the third one of multi-layer) (see Supplementary Movie 1). **d** STEM images for $\alpha = 0°$ (scale bar: 10 nm) (diameter on the x axis: $R_x$, diameter on the y axis: $R_y$). **e** The high-resolution image of a selected area (scale bar: 5 nm), which represent the circular layers of spherical structures of red OLM-TADF-CQDs of size $5 \pm 0.5$ nm in the height curve. **f** Layers and height curves exhibiting regularly arranged structures with an average distance of $0.36 \pm 0.05$ nm between the two layers, which match the model of π–π interaction between CQDs and perylene-like and between perylene-like and perylene-like molecules.

route of monomer CQDs. Figure 1b presents the schematic of assembly process of OLM-TADF-CQDs, in which Fig. 1c corresponds to the structural model of OLM-TADF-CQDs.

Owing to weak interactions between the monomer CQDs, the OLM-TADF-CQDs tend to overlap during the growth process and gradually separate with the increase in the core size of the monomer CQDs (Supplementary Fig. 1). In comparison with the blue, green, yellow, and orange emission OLM-TADF-CQDs, the red OLM-TADF-CQDs exhibit a relatively clear separation between individual OLM-TADF-CQDs. Therefore, the structural evolution associated with red OLM-TADF-CQDs is imaged in 3D using spherical aberration-corrected scanning transmission electron microscope (STEM) tomography. The 3D reconstruction is realized by recording the STEM tilt series of red OLM-TADF-CQDs (tilt angle: $\alpha$ from −70° to +70° in Supplementary Fig. 2) (Supplementary Movie 1)[48–50]. Figure 2a–c and Supplementary Fig. 3 depict the 3D models and reconstructed images, starting from multi-layer (ML) $m_1$ (the first one), $m_2$ (the second one), $m_3$ (the third one) to $m_n$ (the nth one) that are presented to better visualize with false color. The morphology of red OLM-TADF-CQDs at different tilt angles exhibited no significant change. However, the particle size increases when layers are added (circle with radii of 6.0–8.0 nm), which is consistent with the 3D-STEM images. Further observation of the inner structure in a selected area clearly indicates the multi-layered structure of OLM-TADF-CQDs (the insets of Fig. 2a–c). Additionally, the interfacial overlap between each layer (white dashed lines in the inset images) further verifies the packing pattern of monomer CQDs in

the 3D stereotaxic space, which completely differs from previous CQDs. For the STEM image of red OLM-TADF-CQDs at $\alpha = 0°$, the high-resolution image of a selected area (Fig. 2d) exhibits onion-like structures (diameter on the x axis, $R_x = 8.0 \pm 0.20$ nm, and diameter on the y axis, $R_y = 7.5 \pm 0.20$ nm), which are comprised of a polyhedral multi-layered architecture with a high degree of order in a closed nested structure. A selected area is scanned to obtain the end-to-end polyhedral arrangement of several monomer CQDs, similar to the layer–layer configuration shown in Fig. 2e. Moreover, the layered periodic arrangement with an average angle close to 120° (white dashed lines in Fig. 2f) and a distance of $0.36 \pm 0.10$ nm (height curves in Fig. 2f) correspond well with the interlayer structure originating from weak π–π interactions.

## Optical properties

The ultraviolet–visible (UV–vis) absorption spectra of the OLM-TADF-CQDs each exhibit one major absorption peak, centered at 405 nm (blue), 456 nm (green), 484 nm (yellow), 533 nm (orange), and 576 nm (red), with a slight redshift of approximately 10 nm from that of the solution to the solid state (Supplementary Fig. 4). The time-resolved photoluminescence (PL) spectra of OLM-TADF-CQDs measured in the solid state at 77 K (Fig. 3a) exhibit a redshift of the PL peak ($\lambda$) from blue to red between prompt and delayed emissions from 430 nm, 480 nm, 505 nm, 550 nm, and 592 nm at 2 ns to 450 nm, 500 nm, 525 nm, 566 nm, and 611 nm at 10 μs, respectively. Compared to the PL peaks measured at 77 K, those at 300 K exhibit a smaller redshift of less than

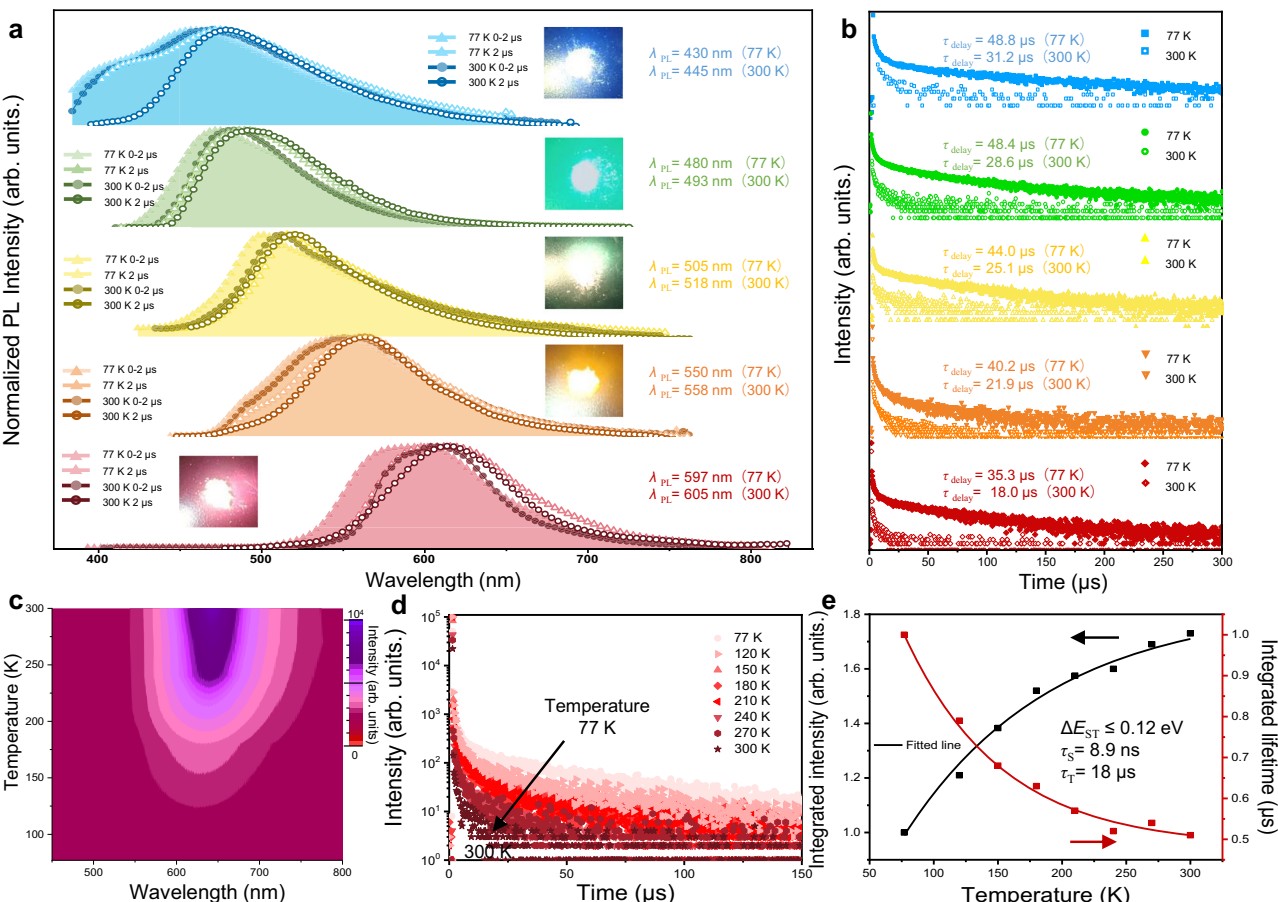

**Fig. 3 | Photophysical properties of the OLM-TADF-CQDs. a** Prompt (<2 ns) and delayed (2 μs) PL spectra at 300 K and 77 K from blue to red emissions (insets indicate the photographs of solid OLM-TADF-CQDs under ultraviolet (UV) light). **b** Time-resolved decay spectra of OLM-TADF-CQDs from blue to red. **c** Two-dimensional pseudocolor map of photoluminescence (PL) spectra of red emission as a function of both temperature and probe wavelength, in which the color scale represents the change of intensity of PL spectra. **d** Temperature-dependent nanosecond–microsecond PL kinetics of the red emission. **e** The plots of the integrated PL emission intensity and the lifetime of red OLM-TADF-CQDs as a function of temperature (77–300 K) ($\Delta E_{ST}$: energy gap between singlet and triplet states).

10 nm. The spectral redshift and emission peak of OLM-TADF-CQDs may be affected by the specific structure of monomer CQDs and the self-assembly between monomers through weak interaction, which is different from the reported CQDs[18,20]. Moreover, emission peaks of the five colors are independent of excitation wavelength (Supplementary Fig. 5). This indicates that emission is characterized by band-edge exciton-state decay rather than defect-state decay owing to the quantum confinement effect, further demonstrating the evident bandgap emission in OLM-TADF-CQDs (Supplementary Figs. 6–8 and Supplementary Table 1). The intensity of PL emission from blue to red is further investigated, where a high absolute QY is found to be 42.3%, 59.5%, 50.0%, 61.0%, and 44.2% in o-dichlorobenzene solution. The QY of OLM-TADF-CQDs in the solid state remains high at 32.0%, 44.7%, 38.0%, 45.0%, and 33.5%, exhibiting only small changes. This is a vast improvement on previously reported CQDs, which show complete quenching of the PL emission intensity owing to the nonradiative energy transfer due to uncontrolled aggregation of the particles.

To further investigate the emission mechanism of TADF, we measure the time-resolved decay spectra of OLM-TADF-CQDs from blue to red. All compounds exhibit lifetimes on the order of microseconds. As indicated in Fig. 3b and Supplementary Fig. 9, the blue, green, yellow, orange, and red emissions have short lifetimes ($\tau$) of approximately 31.2 μs, 28.6 μs, 25.1 μs, 21.9 μs, and 18.0 μs, respectively, under ambient conditions. These emissions experience the RISC process, which is attributed to the delayed emissive species.

Nevertheless, the emission differs slightly from those of previously reported CQDs with long triplet lifetimes at the second scale. Figure 3c illustrate the 2D pseudo-color maps of the intensity change in the PL spectra as a function of both the temperature and probe wavelength. Using the red OLM-TADF-CQDs at 625 nm as an example, the PL intensity gradually increases with increasing temperature, indicating intensity enhancement of the delayed fluorescence under thermal activation. Therefore, the PL emission dependance on temperature specifically validates the formation of thermally activated excitons. Temperature-dependent time-resolved decay spectra further scrutinize the nature of delayed fluorescence, which is strongly temperature-dependent. The lifetimes of OLM-TADF-CQDs decrease with increasing temperature (Fig. 3d and Supplementary Figs. 10 and 11), revealing that excitons present more efficient ISC between triplet and singlet energy levels at higher temperatures[51,52]. Moreover, for the transient absorption (TA) spectra (Supplementary Fig. 12), the negative signals at 500–700 nm with delay times from −0.05 ns to 5 ns and a biexponential decay by TA kinetic traces further verify the TADF characteristics[52]. The small $\Delta E_{ST}$ could be obtained, which are estimated to be 0.12 eV, 0.10 eV, 0.09 eV, 0.06 eV, and 0.06 eV for blue, green, yellow, orange, and red emission, respectively.

Moreover, according to the transient PL decay and QY, the maximum rate constants $k_{ISC}$ and $k_{RISC}$ are up to approximately 11.595 × $10^7$ s$^{-1}$ and 3.777 × $10^5$ s$^{-1}$, respectively (Supplementary Table 2). The large $k_{RISC}$ plays a key role in shortening the lifetime, promoting strong

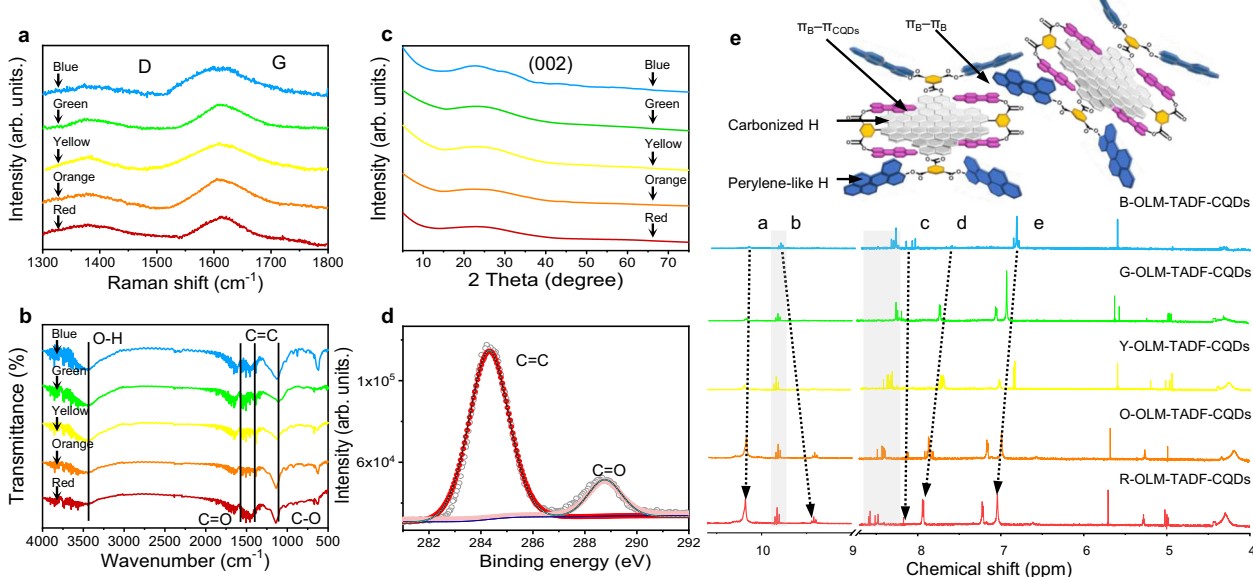

**Fig. 4 | Structural characterizations of the OLM-TADF-CQDs. a** Raman spectra. **b** XRD patterns. **c** FT-IR spectra. **d** C1s spectra. **e** [1]H-NMR spectra from blue (B-), green (G-), yellow (Y-), orange (O-) to red (R-) onion-like multicolor thermally

activated delayed fluorescence carbon quantum dots (OLM-TADF-CQDs) (inset is the partial structure model for OLM-TADF-CQDs, $\pi_B$: the perylene-like molecules, $\pi_{CQDs}$: the carbon core)).

TADF emission. The OLM-TADF-CQDs from blue to red emission exhibit relatively large exciton binding energies of 470 meV, 429 meV, 371 meV, 320 meV, and 280 meV, which are higher than those of several inorganic QDs (Fig. 3e). The high exciton binding energy is ascribed to fewer nonradiative trapping centers, which favors the short lifetime of OLM-TADF-CQDs[18].

Control experiments are performed to further verify the interaction of the assembled structure and monomer CQDs. When the OLM-TADF-CQDs are hydrolyzed by sulfuric acid under refluxing conditions, only weak blue, green, yellow, orange, and red emissions are observed in solution state under UV lamp (Supplementary Fig. 13). In the solid state, all of emissions are almost completely quenched. In contrast to the onion-like structure, the TEM images of these hydrolyzed samples exhibit only single CQDs with a size of 2.5–5.0 nm, indicating the breaking of π–π interactions (Supplementary Fig. 13). Moreover, the functional groups are investigated, and these are almost consistent in monomer CQDs of different colors (Supplementary Fig. 14 and Supplementary Table 3). The specific spatial structure (Supplementary Fig. 15) of monomer CQDs can further promote the assembly of the 3D onion-like structure. The similarity in their chemical compositions indicates that the differences in PL emission of OLM-TADF-CQDs might be mainly related to the different sizes of monomer CQDs. Therefore, the obtained results verify that the blue, green, yellow, orange, and red emissions of OLM-TADF-CQDs originate from the TADF mechanism. We deduce that the triplet state of OLM-TADF-CQDs is mainly stabilized by the assembled structure, which is different from that of monomer CQDs with only weak fluorescence emission.

## Structural characterizations

In order to observe the crystalline structure of monomer CQDs, Raman spectra are collected. Here, the crystalline G band is at about 1615 cm$^{-1}$, and the disordered D band is at about 1380 cm$^{-1}$, in which the G to D intensity ratios ($I_G/I_D$) are at the range of 1.7–2.1 (Fig. 4a and Supplementary Fig. 16). Additionally, the powder X-ray diffraction (XRD) pattern displays a narrow peak centered at approximately 24°, corresponding to (002) crystal face (Fig. 4b and Supplementary Fig. 16 and Supplementary Table 4). Fourier transform infrared (FT-IR) spectra

(Fig. 4c and Supplementary Fig. 16) are observed for the functional groups of OLM-TADF-CQDs. The broad absorption peak in the region of 3000–3600 cm$^{-1}$ corresponds to the stretching of aromatic O–H and C–H bonds. The peaks in 1630 cm$^{-1}$ and 1240 cm$^{-1}$ are associated to the stretching vibration of C=O and C–O, respectively. The X-ray photoelectron spectra (XPS) demonstrate that the OLM-TADF-CQDs are primarily comprised of carbon (C) and oxygen (O) (Supplementary Fig. 17 and Supplementary Table 5). The C=O (288.8 eV) and C=C (284.5 eV) are observed clearly in Fig. 4d. The functional groups further confirm the optimized structural models of monomer CQDs[53,54].

Nuclear magnetic resonance (NMR) experiments are performed for considering the fast proton exchange of free and bound molecules. In the [1]H-NMR spectra (dimethyl sulfoxide (DMSO)-$d_6$, ppm) (Fig. 4e), the signals in the ranges of 9–11 ppm and 6–9 ppm correspond to the hydrogen (H) of perylene-like molecules and carbonized H of the carbon cores, respectively. A signal shift is observed when the five spectra are compared, which is considered to reflect the H–H interaction. The proton signals of the perylene-like substituents (protons a and b in Fig. 4e) undergo a large shift upon complexation with the protons of the carbon cores (protons c–e in Fig. 4e). Moreover, [13]C-NMR spectra (DMSO-$d_6$, ppm) (Supplementary Fig. 18) further confirm the interaction in the assembled structure based on the apparent changes of resonance signals in the range of 162–178 ppm. A signal shift of approximately −3.4 ppm is observed, which is considered to reflect the C–C interaction with different structural assemblies between the $\pi_{CQDs}$ and $\pi_B$. Numerous signals in the range of 120–145 ppm indicate the formation of $sp^2$ domains in the CQDs. These results verify the existence of weak π–π conjugated interactions between monomer CQDs among the 3D onion-like structures.

To further evaluate the assembly process of the 3D onion-like structure, the red OLM-TADF-CQDs are treated with concentrated sulfuric acid for different intervals. The TEM images specifically display the change in structure (Supplementary Fig. 19). After treatment with concentrated sulfuric acid (98%) for 2 h, the structure of the monomer CQDs is displayed in wide-area TEM images, which exhibits an average size distribution of 5 ± 0.3 nm. The high-resolution TEM (HRTEM) images and well-resolved lattice fringe spacing of 0.21 nm corresponding to the (100) interplanar spacing indicate the highly

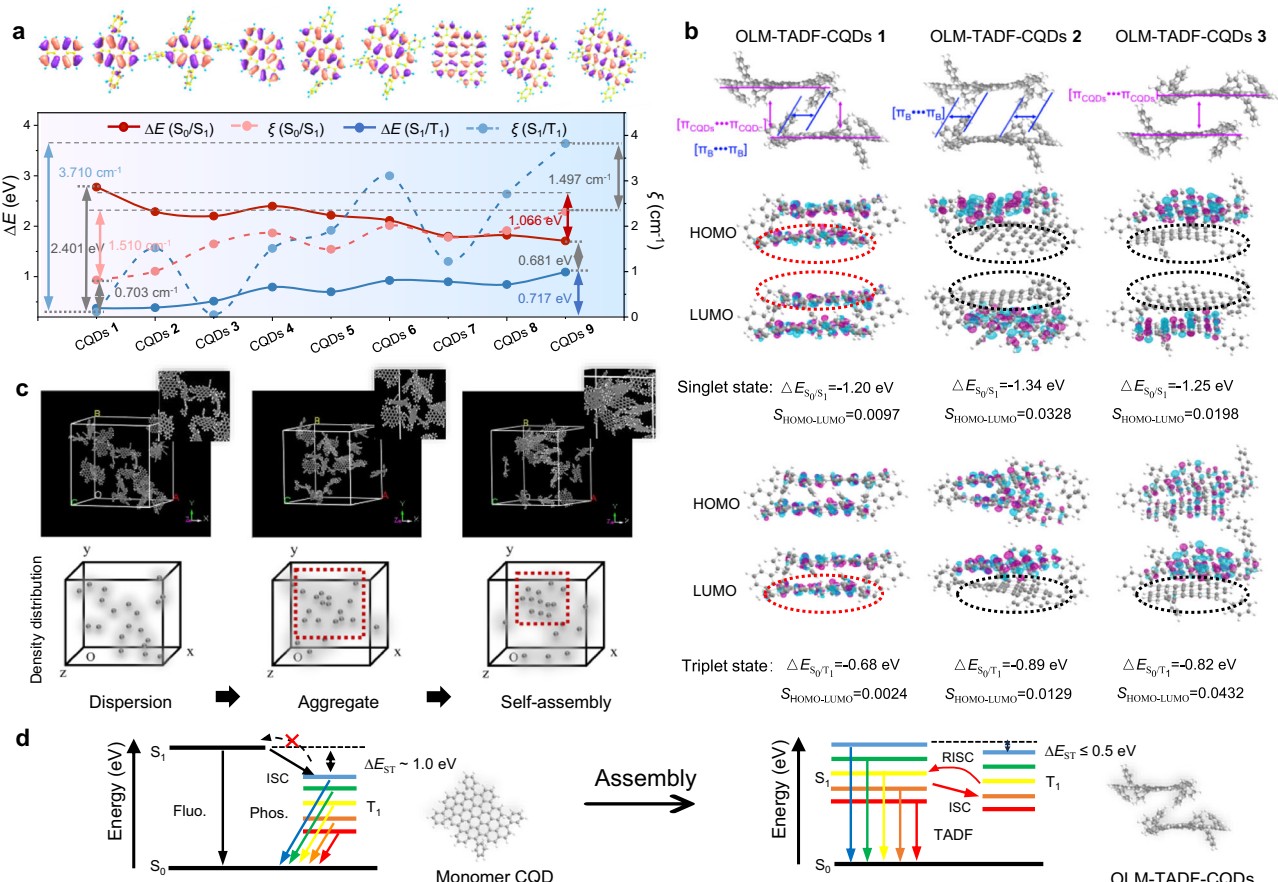

**Fig. 5 | Monomer CQDs calculation and assembly process results. a** Schematics of the calculated energy levels ($E$) and SOC strengths ($\xi$) of $S_0/S_1$ and $S_1/T_1$ of structural models of monomer carbon quantum dots (CQDs) ($S_0$: ground state, $S_1$: singlet excited state, and $T_1$: triplet excited state). **b** Schematics of calculated energy gap ($\Delta E$) and orbital overlap ($S$) of assembled structure models of onion-like multicolor thermally activated delayed fluorescence carbon quantum dots (OLM-TADF-CQDs) using two CQD monomers with $\pi_B$–$\pi_B$ ($\pi_B$: the perylene-like molecules) and $\pi_{CQDs}$–$\pi_{CQDs}$ ($\pi_{CQDs}$: the carbon core), $\pi_B$–$\pi_B$, and $\pi_{CQDs}$–$\pi_{CQDs}$ in the

singlet and triplet states (HOMO: highest occupied molecular orbital, LUMO: lowest unoccupied molecular orbital). **c** Molecular dynamics (MD) simulation based on two structural units: CQDs **7** (CQDs without modification) and CQDs **9** (monomer CQDs with four benzene molecules), changing from dispersion, aggregate, to self-assembly. **d** Schematic of the optical processes of monomer CQDs and assembled structure between monomer CQDs (ISC: intersystem crossing, RISC: reverse intersystem crossing, fluo.: fluorescence emission, phos.: phosphorescence emission, TADF: thermally activated delayed fluorescence).

crystalline graphene-like structure of monomer CQDs. As the concentration of sulfuric acid decreased from 98% to 58% at different intervals (2–0.5 h), a single-layered structure, a two-layered structure, and a semi-circular structure are clearly observed in the wide-area TEM and HRTEM images.

Based on structural characterizations, the assembly process of the 3D onion-like structure is revealed. Monomer CQDs exhibit a large planar $\pi_{CQDs}$ and specific functional groups, which are stacked through weak conjugated interactions, as illustrated in Fig. 1. Considering the three symmetrical chemical bonds in BTC acid ($\theta = 120°$), $\pi_B$ at the end of the $\pi_{CQDs}$ are stabilized at an angle of 120°. The other type of $\pi_B$ is parallel to the $\pi_{CQDs}$ through $\pi$-conjugation ($\theta = 180°$). This is because BTC acid is vertically connected to the $\pi_{CQDs}$ via a rotatable single bond. Furthermore, one monomer CQD is stacked in a staggered pattern with adjacent CQDs according to the weak intermolecular interactions between $\pi_{CQDs}$ and $\pi_B$, thereby twisting to form a corner at approximately 120°. With the continuous stacking between monomer CQDs, a hexagon ring is assembled, leading to the formation of mono-layers and 3D onion-like structures (Fig. 1a).

## Theoretical investigation
Theoretical investigations further verify the indispensability of the spatially twisted structure of monomer CQDs and interaction between

monomer CQDs for the TADF property and self-assembly behavior. From the nine optimized models for monomer CQDs, the HOMO and LUMO energy levels ($E$) and SOC strengths ($\xi$) are calculated for comparison (Supplementary Figs. 20, 21, Supplementary Table 6, and Supplementary Data 1)[44,45]. From CQDs **1** to CQDs **9**, $\xi$ ($S_1$, $T_1$) significantly increases, indicating that benzene molecules around the CQDs can regulate the energy level for different color emission (Fig. 5a and Supplementary Fig. 22). However, the large $\Delta E_{ST}$ of approximately 1.0 eV indicates that the monomer CQDs show the limited contribution to triplet emission, which could scarcely achieve TADF emission, even with specific surface modifications. Figure 5b further depicts three models with different configurations of $\pi$-conjugated interactions between CQDs and perylene molecules of two CQDs monomers for comparing the HOMO and LUMO and orbital overlap coefficient ($S$) of the singlet and triplet states that represents the integral overlap of the electron cloud density distributions of HOMO and LUMO (Supplementary Fig. 23 and Supplementary Data 2)[55,56]. The electron cloud densities in the HOMO and LUMO of OLM-TADF-CQDs **1** are evenly distributed over the two CQD monomers, whereas those of OLM-TADF-CQDs **2** and **3** are predominantly localized on the monomer CQDs with weak extensions to the neighboring ones. Supplementary Table 7 summarizes the changes in $E$ and $S$. OLM-TADF-CQDs **1** has the smallest $\Delta E$ ($S_0/S_1$) of 1.20 eV and $S_{HOMO-LUMO}$ of 0.0097 in the singlet state; in the triplet state, the smallest $\Delta E$ ($S_0/T_1$) of 0.68 eV

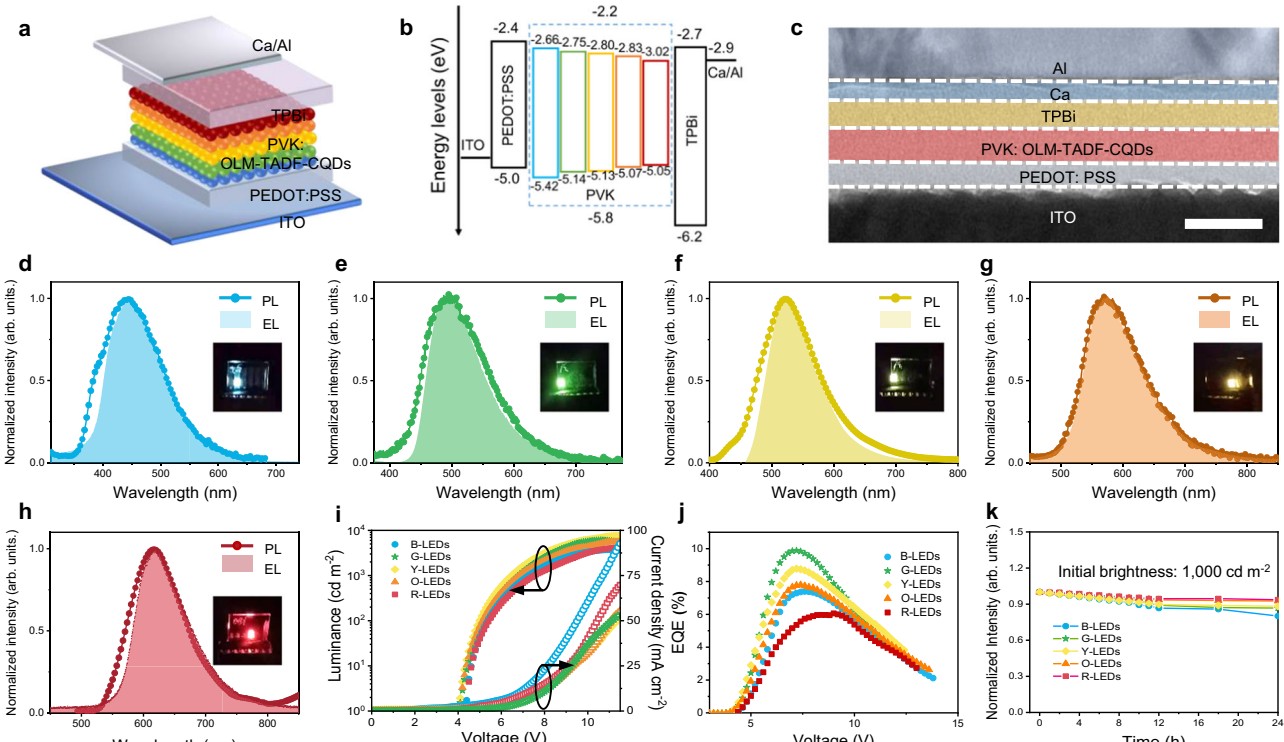

**Fig. 6 | Structure and performance characterization of LEDs. a** Device structure (ITO: Indium tin oxide, PEDOT:PSS: poly(3,4-ethylenedioxythiophene):poly(styrene-sulfonate), PVK: poly(N-vinyl carbazole), OLM-TADF-CQDs: onion-like multicolor thermally activated delayed fluorescence carbon quantum dots, TPBi: 1,3,5-tris(N-phenylbenzimidazol-2-yl) benzene). **b** Energy level diagram. **c** Cross-sectional TEM image (scale bar: 50 nm). **d–h** Electroluminescence (EL) and photoluminescence (PL) spectra of the LEDs at different bias voltages (insets are photographs of the operation of the LEDs). **i** Current density–voltage–luminance (J–V–L) characteristics of LEDs. **j** External quantum efficiency (EQE)–V characteristics of LEDs. **k** Stability plot of LEDs.

and $S_{HOMO\text{-}LUMO}$ of 0.0024 are observed, resulting in a small $\Delta E_{ST}$ of approximately 0.50 eV. In contrast, OLM-TADF-CQDs **2** and **3** have larger $\Delta E_{ST}$ of 0.89 eV and 0.82 eV and $S_{HOMO\text{-}LUMO}$ of 0.0328–0.0129 and 0.0198–0.0432, respectively. The change in $E$ and orbital overlap indicate that OLM-TADF-CQDs **1** from the assembled CQD–CQD with both $\pi_B$–$\pi_B$ and $\pi_{CQDs}$–$\pi_{CQDs}$ can significantly reduce $\Delta E_{ST}$ compared with monomer CQDs[52,57–59]. The spatial structure of monomer CQDs and the specific stacking between monomer CQDs with intermolecular interactions are expected to decrease the orbital overlap for a small $\Delta E_{ST}$, thus producing strong TADF.

To track full assembly pathways of multiple monomer CQDs, molecular dynamics (MD) simulation is conducted to understand the molecular interactions during the self-assembly process of OLM-TADF-CQDs (Fig. 5c)[60,61]. CQDs **9**, a monomer CQDs with four benzene molecules, is selected by considering both the π-conjugation interactions from the carbon plane and benzene molecules and the steric hindrance from the rotation angle of the benzene molecules to the carbon conjugate planes. CQDs **9** monomers prefer to aggregate with each other and then self-assemble in a high-density distribution, stabilizing in the lowest energy state. However, CQDs **7** as a monomer CQDs without modification that features only π-conjugated interactions, and CQDs **8** that with partially modification but without the specific spatial structure, remain in a dispersed state during the simulation process (Supplementary Fig. 24). The present model suitably describes the structural specificity of monomer CQDs and synergetic interactions between the attraction from weak π-conjugated interaction and repulsion from the steric hindrance of surrounding functional groups. Therefore, structural control of monomer CQDs to balance the molecular interaction plays a dominant role in the self-assembly process of OLM-TADF-CQDs (Fig. 5d).

## Electroluminescent devices

The efficient TADF emission from blue to red, adequate solution processability, and high stability render the proposed OLM-TADF-CQDs potential candidates for the development of next-generation display technology. Fabricated thin films of OLM-TADF-CQDs (approximately 25 nm thick) under ambient conditions display efficient multicolor emission with a smooth and uniform surface morphology (Supplementary Figs. 25 and 26). The high-resolution atomic force microscopy image (AFM) of the films further indicates the regular and tightly packed arrangement of the onion-like structure (Supplementary Fig. 27), exhibiting high stability when exposed to oxygen in the air and high temperatures (Supplementary Fig. 28). The device structure is composed of an indium tin oxide (ITO) glass substrate anode, a poly (3,4-ethylenedioxythiophene): poly (styrenesulfonate) (PEDOT:PSS) hole injection layer (HIL), an active poly(N-vinyl carbazole) (PVK)/OLM-TADF-CQDs emission layer, a 1,3,5-tris(N-phenylbenzimidazol-2-yl) benzene (TPBi) electron transport layer (ETL), and a Ca/Al double-layered cathode (Fig. 6a). As indicated in Fig. 6b, the HOMO and LUMO of the active-layer-OLM-TADF-CQDs from blue to red are at 5.80 eV and 2.80 eV, 5.42 eV and 2.66 eV, 5.14 eV and 2.75 eV, 5.13 eV and 2.80 eV, 5.07 eV and 2.83 eV, and 5.05 eV and 3.02 eV, respectively, and the energy barrier is small for the charge injection between different electrodes and OLM-TADF-CQDs (Supplementary Fig. 6). The cross-sectional TEM images further indicate the device structure (Fig. 6c and Supplementary Fig. 29). The corresponding Commission Internationale de l'Eclairage (CIE) coordinates of the blue to red emission are (0.1745, 0.194), (0.2653, 0.4276), (0.2976, 0.4959), (0.4865, 0.4994), and (0.622, 0.3747), at a bias voltage of 4.5 V (Fig. 6c and Supplementary Fig. 30).

Figure 6d–h illustrate the electroluminescence (EL) spectra of the OLM-TADF-CQD-based LEDs. The EL peaks from blue to red are at

**Table 1 | Key optical parameters of OLM-TADF-CQDs for electroluminescent LEDs**

| LEDs | Quantum yield (%) | PL/EL peak (nm) | $V_{on}$ (V) | $L_{max}$ (cd m$^{-2}$) | EQE (%) at $L_{max}$ | $L$ (cd m$^{-2}$) at EQE$_{max}$ | EQE$_{max}$ (%) |
|------|-------------------|-----------------|--------------|--------------------------|----------------------|-----------------------------------|------------------|
| B-LEDs | 41 | 440/444 | 4.3 | 3785 | 4.2 | 1255 | 7.2 |
| G-LEDs | 44 | 490/494 | 4.0 | 6130 | 5.6 | 1400 | 9.9 |
| Y-LEDs | 47 | 520/523 | 4.0 | 7550 | 5.0 | 1900 | 8.7 |
| O-LEDs | 59 | 570/574 | 4.1 | 5580 | 4.5 | 1476 | 7.8 |
| R-LEDs | 45 | 616/620 | 4.2 | 4190 | 3.9 | 1700 | 6.0 |

444 nm, 494 nm, 523 nm, 574 nm, and 620 nm. The EL peaks correspond well with the measured PL peaks, indicating that the emission originates from the OLM-TADF-CQDs (Supplementary Fig. 31). The apparent voltage-independent emission color demonstrates the high color stability of the LEDs. The photographs of the operations of OLM-TADF-CQD-based LEDs (insets in Fig. 6d–h) display the bright emission, further indicating the high color stability of the LEDs. Figure 6i depicts the luminance and current density of OLM-TADF-CQD-based LEDs under different voltage. The turn-on voltages ($V_{on}$) of OLM-TADF-CQD-based LEDs are at the range of 4.0 V to 4.3 V. The low $V_{on}$ are from the matching energy levels of OLM-TADF-CQD-based LEDs[20]. Table 1 summarizes the $L_{max}$ and current efficiency ($\eta_c$) of the LEDs from blue to red. The device performance of OLM-TADF-CQDs is considerably improved, which is predominantly due to the utilization of triplet excitons.

Significantly, the maximum EQEs of the OLM-TADF-CQD-based devices are 6.0–9.9% (Fig. 6j and Supplementary Fig. 32). Such high EQE values signify the utilization of triplet excitons, surpassing the efficiency limits of conventional electroluminescent LEDs based on fluorescence emission. A device with typical EQE of 5% harvests merely 25% of singlet excitons. This indicates that TADF has a critical role in boosting device performance, where triplet excitons could efficiently emit strong delayed fluorescence owing to the formation of the 3D onion-like structure assembled by monomer CQDs. The nearly 10% EQE verifies the extensive potential of OLM-TADF-CQDs as an active emission layer for the development of CQD-based electroluminescent LEDs. As an ultra-stable feature of TADF emission, the high-efficiency OLM-TADF-CQD-based LEDs exhibit good ambient stability compared to previous reports[18,20]. After operation for 24 h, more than 85% of the initial luminance ($L_0 = 1000$ cd m$^{-2}$) is retained (Fig. 6k). The high stability further verifies the potential applications of OLM-TADF-CQDs for the development of high-performance LEDs.

### Flexible device characterization and performance

Optoelectronic devices are increasingly becoming of interest for commercial applications[62–64]. Bright and efficient LEDs with red emission are highly desirable as one of the key optoelectronic components for realizing full-color displays and lighting. Considering the high efficiency and stability of the red OLM-TADF-CQDs spun on ITO-coated glass substrates, substantial scope exists for the advancement of flexible LEDs. Supplementary Fig. 33 depicts the strain distribution of the flexible OLM-TADF-CQD thin film obtained by spin-coating OLM-TADF-CQDs on a polyethylene glycol terephthalate (PET) substrate. The bending strain predominantly occurs in the center of the film with tested bending radii ($r$) of ∞ and 2.5 mm, exhibiting the flexibility of OLM-TADF-CQD thin films owing to the weak van der Waals bonds between molecules. The PL mapping of the OLM-TADF-CQD thin films under different $r$ indicated that cracks begin to appear when the bending radius reached approximately 2.5 mm. The corresponding SEM images also verify the onset of fractures at a $r$ of approximately 2.5 mm, which are observed in the SEM images with the root-mean-square roughness varying from 2.41 nm to 5.63 nm (Supplementary Fig. 33).

The flexible device structure is composed of an ITO PET substrate anode, PEDOT:PSS as the HIL, an active PVK/red OLM-TADF-CQD emission layer, a TPBi as the ETL, and a Ca/Al double-layered cathode.

Photographs of the red OLM-TADF-CQD-based LEDs (Fig. 7a) indicate a close-up view of the bright, uniform, and defect-free surface EL. The EL peaks of the flexible red OLM-TADF-CQD-based LEDs are located at 621 nm (Fig. 7b), corresponding well to that in glass substrate. The emission is independent of voltage, indicating the high color stability of the LEDs. Figure 7c depicts the luminance and current density curves of the red OLM-TADF-CQD-based LEDs under different voltage. The $V_{on}$ is approximately 4.5 V. And $L_{max}$ and $\eta_c$ of the flexible LEDs reach approximately 2554 cd m$^{-2}$ and 1.05 cd A$^{-1}$, respectively. After operating for 300 s, more than 99% of the initial luminance ($L_0 = 1000$ cd m$^{-2}$) is retained (Fig. 7d). In contrast to LEDs based on a hard glass substrate, flexible device performance is substantially affected by the uncontrolled morphology and thinness of the film on the PET substrate. Notably, the flexible electroluminescent LEDs based on red OLM-TADF-CQDs are realized, indicating a promising future for commercial standardization of flexible LED displays. This can pave the way for applying CQDs in high-performance optoelectronics.

### Discussion

In this study, we report the fabrication of OLM-TADF-CQDs with multicolor emissions by assembling monomer CQDs into a 3D onion-like structure. The synthesis is performed by selecting 1,3,5-BTC acid and 3,4,9,10-perylenetetracarboxylic dianhydride as precursors to generate multicolor TADF from blue to red emission with QYs of 42.3–61.0%. Detailed structural and optical characterizations and theoretical calculations reveal that the monomer CQDs with specific structure could regulate SOC and energy levels of the singlet state and triplet states for full-color emission. Moreover, the strategically assembled structure of monomer CQDs is expected to generate TADF emission by stabilizing the triplet state and reducing the orbital overlap to decrease $\Delta E_{ST}$. The LEDs based on the OLM-TADF-CQDs display high stability and performance with $L_{max}$ of 3785–7550 cd m$^{-2}$ and maximum EQEs of 6.0–9.9%. Furthermore, the flexible LEDs exhibit red emission with a $L_{max}$ of 2554 cd m$^{-2}$. Delayed fluorescence of CQDs is achieved through the elaborate structural design by stepping from 0D to 2D and finally to 3D, and are also successfully used in electroluminescent LEDs with high device performance. An EQE of nearly 10% symbolizes milestone progress of CQDs for next-generation high-performance displays. We anticipate that the findings of this study will stimulate more research interests in the design of CQDs and device optimization based on CQDs.

### Methods

#### Synthesis of the onion-liked structure

OLM-TADF-CQDs are synthesized by solvothermal treatment, in which 1,3,5-benzenetricarboxylic acid (45 mg, 0.178 mmol) and 3,4,9,10-perylenetetracarboxylic dianhydride (2 mg, 0.0051 mmol) are dissolved in concentrated H$_2$SO$_4$ (1–5 mL), followed by a formamide solution (15 mL). The volume ratios of concentrated sulfuric acid to formamide for the different colors are 1:15 (blue), 1:10 (green), 1:8 (yellow), 1:6 (orange), and 1:5 (red). The precursor solution is ultrasonically dissolved in a poly(tetrafluoroethylene) (Teflon)-lined autoclave (25 mL) for 10 min and then heated to 110 °C (blue), 125 °C (green), 150 °C (yellow), 180 °C (orange), and 200 °C (red). The reaction times are 15 min (blue), 0.5 h (green), 1 h (yellow), 2 h (orange), and 4 h (red).

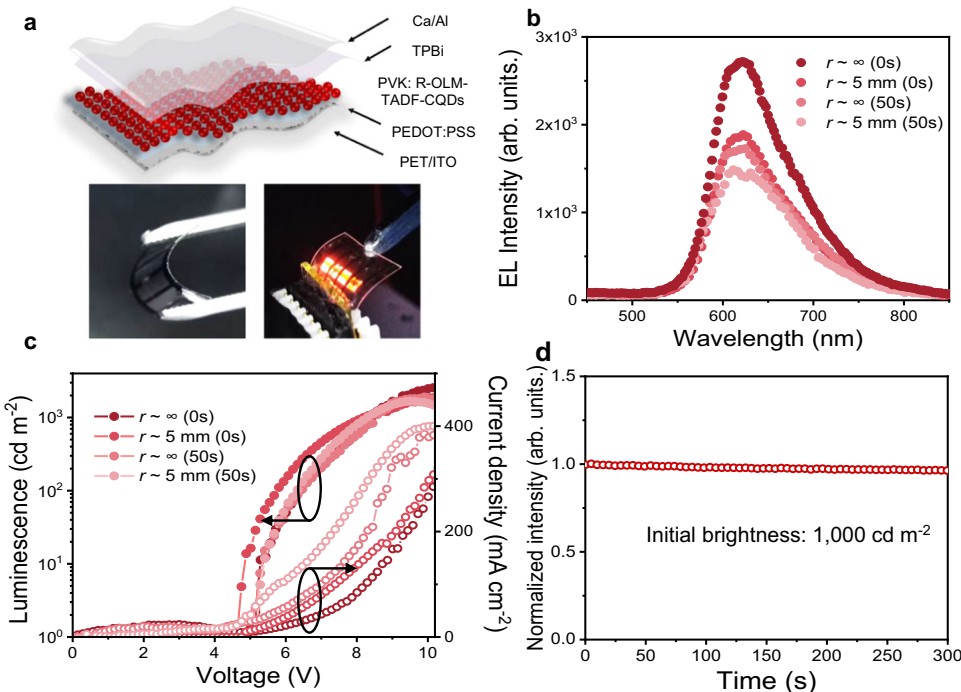

**Fig. 7 | Performance characterization of R-OLM-TADF-CQD film and flexible LEDs. a** Device structure and photographs of the flexible LEDs based on red OLM-TADF-CQDs (PET: Polyethylene terephthalate, ITO: Indium tin oxide, PEDOT:PSS: poly(3,4-ethylenedioxythiophene):poly(styrene-sulfonate), PVK: poly(N-vinyl carbazole), R-OLM-TADF-CQDs: red onion-like multicolor thermally activated delayed fluorescence carbon quantum dots, TPBi: 1,3,5-tris(N-phenylbenzimidazol-2-yl) benzene. **b** Electroluminescence (EL) spectra of the LEDs at different bias voltages (*r*: bending radii). **c** *J–V–L* characteristics of LEDs. **d** Stability plot of LEDs.

The autoclave then cools to room temperature naturally (25 °C). The crude product solution is neutralized to pH = 7 by centrifugation and washes three times with $Na_2CO_3$ solution (0.01 mol L$^{-1}$), and the obtained products are repeatedly extracted and purified by dichloromethane. After drying, the OLM-TADF-CQDs are separated twice through silica gel column chromatography by changing the volume ratio (*v/v*) of petroleum ether: ethyl acetate = 1:1 for all crude products in the first separation; 1:6 (blue), 1:7 (green), 2:15 (yellow), 1:9 (orange), and 1:10 (red) in the second separation. The isolated masses of these five samples are approximately 1–2 mg, keeping the 3–5% yields. Note that the solid samples cannot been obtained at high temperature to avoid clumping and deterioration.

## Synthesis of the monomer carbon quantum dots

Monomer CQDs are obtained by hydrolyzed the OLM-TADF-CQDs (0.5 mg) using $H_2SO_4$ (5 mL, 30 % in water). In a glove box, the reaction mixture is added in a dry 15-mL vial equipped with a magnetic stirrer bar, and stirred at 50 °C for 36 h and then cooled down to room temperature. Then, $Na_2CO_3$ solution (0.01 mol L$^{-1}$) is used to neutralize the solution and the resulting mixture is extracted with DCM (20 mL for five times). The combined organic layers are dried over $Na_2SO_4$, filtered, concentrated under reduced pressure and purified via column chromatography on silica gel (methanol: dichloromethane = 1:1 for first separation, and ethyl acetate: dichloromethane: triethylamine = 50:5:1 for second separation) with to afford the monomer CQDs.

Synthesis of OLM-TADF-CQDs, quantum yield measurements, ultraviolet photoelectron spectroscopy measurement, characterization method, STEM images characterization, theoretical calculations, as well as device fabrication and characterization are available in detail in Supporting Information.

## Data availability

The authors declare that the data used in this study are available within the article and Supplementary Information file, or from the corresponding author upon request. Coordinates of the optimized structures are available from the Supplementary Data.

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

## Acknowledgements

This work is financially supported by National Natural Science Foundation of China (NSFC 22172008, L.F.; 22372009, Y.Z.; 22302012, F.Y. and 21872010, L.F.), National Key Research and Development Program of China (2019YFE0112200, Z.T. and 2023YFB3611800, F.Y.), Special Funds for Introducing Talents from Beijing Normal University (111032315, F.Y.) and National Science Fund for Excellent Young Scholars (Overseas, F.Y.).

## Author contributions

Y.S., L.F., Y.Z., F.Y. and Z.T. designed the experiment. Y.S. and Z.W. fabricated the LED devices. Y.S., T.Y. and T.M. fabricated the samples for structural characterization. Y.S., Y.Z., Y.L., X.L., Z.T., F.Y. and L.F. analyzed the data. Y.S. and T.Y. performed the theoretical simulations and analyzed the data. Y.S., Y.Z., Z.T., F.Y. and L.F. wrote the paper.

## Competing interests

The authors declare no competing interests.
