## [Peer Review File · Nature Communications]

Onion-like Multicolor Thermally Activated Delayed Fluorescent Carbon Quantum Dots for Efficient Electroluminescent Light-emitting DiodesEditorial Note: This manuscript has been previously reviewed at another journal that is not operating a transparent peer review scheme. This document only contains reviewer comments and rebuttal letters for versions considered at *Nature Communications*.

REVIEWER COMMENTS

Reviewer #2 (Remarks to the Author):

In this resubmitted paper, authors have almostly addressed the reviewers' comments, and the corresponding revisions have been made. I think this paper may be acceptable at this stage for the publication in Nature Communications.

Reviewer #3 (Remarks to the Author):

The authors have thoroughly revised the manuscript as per all three reviewers' suggestions and satisfactorily answered all the comments with necessary changes/additions of data in the revised manuscript. The authors also introduced additional promising studies in the revised manuscript. The uniqueness of the reported onion-like multicolor TADF-CQDs (OLM-TADF-CQDs) is that they are synthesized for the first time with impressive quantum yields of 42.3–61.0%. The revised manuscript with background information and results is quite interesting for LED applications and upcoming research. Hence, I recommend the acceptance of the revised manuscript in the Nature Communication Journal.

Reviewer #4 (Remarks to the Author):

review attached

In their article, Shi et *al.* investigated the optical properties and prepared organic light-emitting diodes (OLEDs) made of carbon quantum dots (CQDs). They conducted a multidisciplinary study which involves beyond the prepared devices, steady-state and time-resolved spectroscopy, STEM, X-Ray spectroscopy, Raman spectroscopy, NMR and theoretical calculations.

I think the amount of experimental data is impressive, but I still think they don't completely support the claims of the authors:

- Although in view of the PLQY's of the different compounds and the EQE's measured, TADF is occurring in these quantum dots, I think there is not proposition on the TADF mechanism. For instance, what are the roles of the CQD core and the perylene substituents? These systems would be perfect for transient absorption experiments.
- In relation to my previous comment. It would be important to have a comment on the broad emission spectra of these materials. CQDs are well defined rigid systems, and I would expect a narrow emission.
- Regarding the theoretical calculations, I find some difficulties to understand what the authors are trying to prove. The CQDs are a self-assembly of the monomer which includes the perylene substituents. In the MD simulations, the carbon cores seem to be omitted and are not representative of the systems studied experimentally.
- The TD-DFT calculations are carried out based on the stacked carbon cores and any computed optical properties cannot be representative of the experimental systems again. The ΔE_{ST} are as expected far from what it is observed experimentally. It seems that the authors want to stress that the ΔE_{ST} and thus TADF is a property of the carbon core which I hardly believe.
- Complementary comment to my previous one: It is difficult to believe in view of the orbitals that the HOMO-LUMO overlap is almost vanishing. There should be a problem there.

Point-by-point responses to Reviewers' comments

We thank the Reviewer for her/his valuable and constructive comments, and have carefully addressed them as following. The Reviewer's comments are provided in black font, and our response to each comment is provided in red font (the corresponding revision to each response is indicated in bold text).

In their article, Shi et al. investigated the optical properties and prepared organic light-emitting diodes (OLEDs) made of carbons quantum dots (CQDs). They conducted a multidisciplinary study which involves beyond the prepared devices, steady-state and time-resolved spectroscopy, STEM, X-Ray spectroscopy, Raman spectroscopy, NMR and theoretical calculations.

We are very appreciative of the Reviewer's recognition of our work.

I think the amount of experimental data is impressive, but I still think they don't completely support the claims of the authors:

We also thank the Reviewer's constructive and valuable comments, which will be addressed carefully point by point as following.

1. Although in view of the PLQY's of the different compounds and the EQE's measured, TADF is occurring in these quantum dots, I think there is not proposition on the TADF mechanism. For instance, what are the roles of the CQD core and the perylene substituents? These systems would be perfect for transient absorption experiments.

Reply: We appreciate the Reviewer's recognition of TADF occurring in OLM-TADF-CQDs.

For the TADF mechanism, we did give the detailed investigation and descriptions in the section of "Optical properties of OLM-TADF-CQDs" and "Theoretical investigation" as well. Together with the control experiments, we concluded that the monomer CQDs with specific structure could modulate the energy levels for full-color emission, and the strategically assembled 3D onion-like structure from monomer CQDs is expected to stabilize the triplet excited state and reduce the orbital overlap to decrease ΔE_{ST} , thus generating TADF emission.

The carbon core and the perylene substituents that exist in the monomer CQDs have two roles: (1) regulate energy levels for different PL emission; (2) generate the specific spatial structure (Fig. S13) that can promote the self-assembly between monomer CQDs for the formation of onion-like structure. We added the corresponding description in the revised manuscript to further clarify it as below.

Moreover, the specific spatial structure (Supplementary Fig. 14) of monomer

CQDs can further promote the assembly of the 3D onion-like structure.

In addition, we followed the Reviewer's suggestion, and carried out the transient absorption measurements of OLM-TADF-CQDs with red emission. As shown in **Supplementary Fig. 11**, the results displayed that the negative signals at 500–700 nm changed with delay times from -0.05 to 5 ns, in which ns-TA spectroscopy was commonly used to observe these states with long-lived (up to μs). And a biexponential decay that was a character of TADF was also observed, in which one component could be ascribed to the prompt fluorescence with lifetime in the range of a few nanosecond (ns), and the other was the delayed fluorescence with lifetime in the microsecond (μs) range.

Supplementary Figure 11. (A) Transient absorption spectra of red OLM-TADF-CQDs with a pump wavelength of 400 nm at indicated delay times. The positive signals at 400–500 nm can be ascribed to the excited-state absorption, and the negative signals at 500–700 nm correspond to the ground state bleaching and stimulate emission. (B) Bleach signal kinetic for $\lambda_{\text{pump}} = 400$ nm and probe wavelength of 580 nm.

Correspondingly, we added the description and reference as **53** in the section of “Optical properties of OLM-TADF-CQDs” in the revised manuscript as below.

Moreover, for the transient absorption (TA) spectra (Supplementary Fig. 11), the negative signals at 500–700 nm with delay times from -0.05 to 5 ns and a biexponential decay by TA kinetic traces further verify the TADF characteristics.⁵³

2. In relation to my previous comment. It would be important to have a comment on the broad emission spectra of these materials. CQDs are well defined rigid systems, and I would expect a narrow emission.

Reply: In fact, due to the strong electron-photon coupling and surface defects, the conventional CQDs commonly exhibit a broad emission with full width at half maximum (FWHM) exceeding 80 nm. The FWHM decreased significantly until the discovery of unique triangular CQDs (~30 nm) in our lab, in which the triangular structural rigidity dramatically reduces electron-phonon coupling, thus giving rise to the free-excitonic emission with negligible trap states (*Nat. Commun.* 2018, 9, 2249). This was further verified by the synthesis of a narrow deep-blue CDs with specific

functional groups (*Nat. Photonics* 2020, 14, 171).

For OLM-TADF-CQDs in this work, the broad emissions are attributed to the following two aspects. (1) As shown in Fig. S13, monomer CQDs are composed of a carbon core and perylene-like molecules as surface functional group, which exhibit a broad emission. (2) Through self-assembly between monomer CQDs with specific spatial structure, OLM-TADF-CQDs with 3D onion-like structure achieve TADF emission, leading to a broad emission spectrum.

In order to clarify it, we revised the description in the revised manuscript as below.

The spectral redshift and emission peak of OLM-TADF-CQDs may be affected by the specific structure of monomer CQDs and the self-assembly between monomers through weak interaction, which is different from the reported CQDs.^{18,20}

3. Regarding the theoretical calculations, I find some difficulties to understand what the authors are trying to prove. The CQDs are a self-assembly of the monomer which includes the perylene substituents. In the MD simulations, the carbon cores seem to be omitted and are not representative of the systems studied experimentally.

Reply: For the theoretical investigations, there are mainly three aspects to prove: (1) the carbon core and surface functional groups in monomer CQDs can regulate energy level, achieving different emission wavelength; (2) the specific stacking between monomer CQDs can stabilize triplet state and decrease ΔE_{ST} , promoting TADF emission; (3) in a high-density distribution, the monomer CQDs with specific spatial structure and functional groups can aggregate with each other, showing a tendency to self-assembly.

In order to further clarify it, we revised the descriptions in the section of “Theoretical investigation” as below:

Benzene molecules around the CQDs can regulate the energy level for different color emission (Fig. 4a and Supplementary Fig. 21). However, the large ΔE_{ST} of approximately 1.0 eV indicates that the monomer CQDs show the limited contribution to triplet emission, which could scarcely achieve TADF emission, even with specific surface modifications.

In MD simulations, the carbon core in these structure models is not omitted. There are two parts, (1) the pure carbon core (CQDs-7 in Fig. S19), (2) the carbon core with specific spatial structure that modified by functional groups (CQDs-8 and CQDs-9 in Fig. S19). These models are designed to correspond well with the actual structure of OLM-TADF-CQDs, including the relative sizes between carbon core and surface functional groups, the number of surface functional groups, the twist angle and weak interaction between carbon core and surface functional groups. Based on this, the

simulated results showed the difference of aggregate trend between carbon core and the carbon core with specific spatial structure, which is consistent with the experimental results of self-assembly process from the monomer CQDs to 3D onion-like structure, and is completely different from the reported CQDs with only dispersed individual dots.

In order to further clarify the structure of monomer CQDs, in revised supplementary information, we highlight the carbon core (**white dotted line**) and functional groups (**red dotted line**), respectively. And we also added a MD simulation of CQDs-8 as **Supplementary Fig. 23B** to further prove the self-assembly process and revised the corresponding description in revised manuscript as below.

CQDs-7 as a monomer CQDs without modification that features only π -conjugated interactions, and CQDs-8 that with partially modification but without the specific spatial structure, remain in a dispersed state during the simulation process.

Supplementary Figure 23. Molecular dynamics (MD) simulation based on two structural units: CQDs-7 (A) CQDs-8 (B) and CQDs-9 (C) (the carbon core is highlighted as white dotted line, and the functional groups are highlighted as red dotted line). Monomer CQDs were constructed in a cubic water box of $50 \times 50 \times 50 \text{ \AA}^3$ and equilibrated by 200 ns of MD simulation. Twenty monomer units each were packed into a confined layer. The constant pressure and temperature ensemble simulations were conducted at 1 atm and 298 K, maintained with the Berendsen barostat and Andersen thermostat, respectively, producing the target density of 0.1 g cm^{-3} .

4. The TD-DFT calculations are carried out based on the stacked carbon cores and any computed optical properties cannot be representative of the experimental systems again. The ΔE_{ST} are as expected far from what it is observed experimentally. It seems that the authors want to stress that the ΔE_{ST} and thus TADF is a property of the carbon core which I hardly believe.

Reply: Considering the large size of the real structures for OLM-TADF-CQDs, all of TD-DFT calculated models (Fig. 4a) are designed by selecting the representative units in the real structure of monomer CQDs (Fig. S13), including (1) the relative sizes of carbon core and surface functional groups, (2) the number of surface functional groups and linking sites on carbon core, (3) the twist angle and weak interaction between carbon core and surface functional groups.

There are two lines for the changing trend of these calculation results to support the experimental systems:

(1) The carbon core and surface functional groups in monomer CQDs can regulate energy level. As shown in Fig. 4a and S20, with the increase of functional groups around carbon core with different size, the energy level of LUMO increase from 1.90 to 2.72 eV, and the energy level of HOMO decrease from 4.74 to 4.13 eV. These calculation results verify the experimental result that the monomer CQDs with different size determines the different emission color of OLM-TADF-CQDs (Fig. S11-13).

(2) The orderly stacking between monomer CQDs can stabilize triplet state and decrease ΔE_{ST} . As shown in Fig. 4b, with the structure from monomer CQDs to orderly stacking between monomer CQDs, ΔE_{ST} decreases from 1.0 eV to 0.5 eV. The change trend of ΔE_{ST} can support the experimental result that OLM-TADF-CQDs assembled by monomer CQDs exhibits TADF emission, but individual monomer CQD only shows weak fluorescence emission.

Therefore, we concluded that monomer CQDs with different sizes show different PL emission, and the 3D onion-like structure assembled by monomer CQDs can regulate ΔE_{ST} to achieve TADF emission.

5. Complementary comment to my previous one: It is difficult to believe in view of the orbitals that the HOMO-LUMO overlap is almost vanishing. There should be a problem there.

Reply: We are very sorry for the misunderstanding that the HOMO-LUMO overlap seems almost vanishing, which was caused by rounding the significant figures in Fig. 4b.

In fact, these calculated $S_{HOMO-LUMO}$ (the integral overlap of the electron clouds of HOMO-LUMO distribution) should be ~ 0.0097 (S) and ~ 0.0024 (T) for OLM-TADF-CQDs-1; ~ 0.0328 (S) and ~ 0.0129 (T) for OLM-TADF-CQDs-2; and ~ 0.0198 (S) and ~ 0.0432 (T) for OLM-TADF-CQDs-3, respectively.

Monomer CQDs are composed with a carbon core and surface functional groups, in which the large steric hindrance and twisting angle ($\sim 90^\circ$) lead to the relatively small

orbital overlap. In addition, the specific orderly stacking of OLM-TADF-CQDs-1 results in a smaller orbital overlap than that of OLM-TADF-CQDs-2 and OLM-TADF-CQDs-3. In order to further clarify it, we relabeled these values in Fig. 4b and Table S4, and revised them in revised manuscript.

REVIEWERS' COMMENTS

Reviewer #2 (Remarks to the Author):

The authors have basically addressed reviewer's comments. Although not perfect, the paper may be acceptable at this stage.

Reviewer #3 (Remarks to the Author):

The author's response to the reviewer's comments demonstrates a comprehensive and meticulous approach to addressing each concern raised. First and foremost, the clarity provided regarding the TADF mechanism is admirable. The detailed explanations in the "Optical properties of OLM-TADF-CQDs" and "Theoretical investigation" sections, along with the incorporation of transient absorption measurements, contribute significantly to establishing the occurrence of TADF in OLM-TADF-CQDs. The experimental evidence presented in Supplementary Fig. 11, showcasing a bi-exponential decay characteristic of TADF, serves as a robust validation of the claims made in the study. This thorough response not only addresses the specific concern raised by the reviewer but also adds depth to the overall understanding of the research. Furthermore, the authors adeptly explain the broad emission spectra observed in OLM-TADF-CQDs. By attributing the broad emissions to the specific structure of monomer CQDs and the self-assembly between monomers, the authors provide a nuanced perspective on the emission behavior. The clarification distinguishes OLM-TADF-CQDs from conventional CQDs, emphasizing the importance of the unique spatial structure in achieving TADF emission. This additional context not only resolves the reviewer's concern but also enhances the overall clarity and depth of the research article. the author's answers to the reviewer's comments are thorough, scientifically sound, and effectively enhance the overall quality of the research article. I recommend this article for the publication.